# Endoscopic Diagnosis of Eosinophilic Esophagitis: Basics and Recent Advances

**DOI:** 10.3390/diagnostics12123202

**Published:** 2022-12-16

**Authors:** Yasuhiko Abe, Yu Sasaki, Makoto Yagi, Naoko Mizumoto, Yusuke Onozato, Matsuki Umehara, Yoshiyuki Ueno

**Affiliations:** 1Division of Endoscopy, Yamagata University Hospital, Yamagata 990-2321, Japan; 2Department of Gastroenterology, Faculty of Medicine, Yamagata University, Yamagata 990-2321, Japan

**Keywords:** eosinophilic esophagitis, endoscopic diagnosis, EREFS scoring, image enhanced endoscopy, diagnostic accuracy

## Abstract

Eosinophilic esophagitis (EoE) is a chronic, immune-mediated inflammatory disease, characterized by esophageal dysfunction and intense eosinophil infiltration localized in the esophagus. In recent decades, EoE has become a growing concern as a major cause of dysphagia and food impaction in adolescents and adults. EoE is a clinicopathological disease for which the histological demonstration of esophageal eosinophilia is essential for diagnosis. Therefore, the recognition of the characteristic endoscopic features with subsequent biopsy are critical for early definitive diagnosis and treatment, in order to prevent complications. Accumulating reports have revealed that EoE has several non-specific characteristic endoscopic findings, such as rings, furrows, white exudates, stricture/narrowing, edema, and crepe-paper esophagus. These findings were recently unified under the EoE endoscopic reference score (EREFS), which has been widely used as an objective, standard measurement for endoscopic EoE assessment. However, the diagnostic consistency of those findings among endoscopists is still inadequate, leading to underdiagnosis or misdiagnosis. Some endoscopic findings suggestive of EoE, such as multiple polypoid lesions, caterpillar sign, ankylosaurus back sign, and tug sign/pull sign, will aid the diagnosis. In addition, image-enhanced endoscopy represented by narrow band imaging, endocytoscopy, and artificial intelligence are expected to render endoscopic diagnosis more efficient and less invasive. This review focuses on suggestions for endoscopic assessment and biopsy, including recent advances in optical technology which may improve the diagnosis of EoE.

## 1. Introduction

Eosinophilic gastrointestinal disorders (EGIDs) are refractory gastrointestinal inflammatory entities that cause chronic eosinophilic inflammation and dysfunction in the gastrointestinal tract based on excessive Th2-dominant immunological responses to dietary antigens [1]. EGIDs are broadly classified into eosinophilic esophagitis (EoE), in which the inflammation is confined to the esophagus, and eosinophilic gastroenteritis (EGE), in which the gastrointestinal tract is widely affected, irrespective of esophageal involvement. EoE has been widely recognized as a major cause of dysphagia and food impaction in adolescents and adults, with a dramatically increasing prevalence worldwide over the past several decades [2,3,4,5].

EoE is diagnosed based on the presence of clinical symptoms and histologically-proven esophageal eosinophilia with a peak of ≥15 eosinophils/hpf (~60 eosinophils/mm^2^), excluding diseases and conditions inducing secondary esophageal eosinophilia such as EGE, hyper eosinophilic syndrome, drug-induced esophagitis or achalasia [6]. Thus, EoE is a clinicopathological disease that requires histopathological evidence of intense intraepithelial eosinophils for a definitive diagnosis. Therefore, it is crucial for diagnosis that endoscopists master EoE’s peculiar endoscopic findings, such as edema, rings, furrows, white exudates and strictures, which represent a main gateway for its identification [7]. The EoE endoscopic reference score (EREFS) is expected to standardize the endoscopic assessment, and thus improve diagnostic accuracy [8]. However, these findings are not always specific to EoE and vary in prevalence, degree, and distribution within the esophagus. More notably, up to 10% of patients with EoE have a normal-appearing esophagus [7]. This is another challenge, represented by the low diagnostic agreement among and within endoscopists. Therefore, despite the development and revisions of clinical guidelines and consensus, endoscopic findings have not yet been incorporated into the diagnostic criteria for EoE [9,10,11,12]. Several efforts to improve recognition and diagnostic accuracy of endoscopic abnormalities using image enhancement endoscopy (IEE), endocytoscopy, and artificial intelligence (AI) have been recently reported. In this article, we review the basics and the most recent advances and challenges in the endoscopic diagnosis of EoE.

## 2. Typical Presentation of EoE with Food Impaction

EoE is also referred to as steak house syndrome, as the most commonly impacted foods are meat products [13,14]. Upon urgent endoscopy for a patient with food impaction, adhesion of food residues on the esophageal mucosa, or reflux of gastric contents into the esophagus, may prevent clear observation by the endoscopist, even after impacted food removal from the esophagus; thus, they may potentially miss EoE diagnosis. The food bolus is successfully removed in the majority of patients at the first endoscopy; however, esophageal biopsies fail to be obtained in up to 40% of the patients undergoing endoscopy for removal of impacted food [13,15,16]. This implies that endoscopists may find it difficult to perform esophageal biopsies in addition to the removal of food bolus in the emergency setting. Other reports found that endoscopy within 6 months to 2 years of initial food impaction was performed only in 45–78% of cases, and esophageal biopsies were obtained only in 28–76% of cases [16,17,18]. To avoid missing EoE, it is critical to perform re-endoscopy in patients with food impaction as soon as possible (Figure 1). The American Society for Gastrointestinal Endoscopy consensus statements recommend that biopsy be performed, if possible, at initial food impaction in suspected EoE, avoiding the impaction site which may be potentially injured [19].

Notably, proton pump inhibitors (PPI) used prior to endoscopic examination can mask EoE endoscopically and histologically [20]. Hillman et al. estimated that EoE was missed in 1 of 5 patients with food impaction biopsied while on a PPI [16]. PPIs may also improve acid reflux symptoms, resulting in a reduction in medical visits and an increase in missed EoE cases. Since short-term recurrence of food impaction is uncommon, less than 5% within 1 year, re-endoscopy following discontinuation of medications is recommended [16]. In addition, endoscopists should consider that esophageal mechanical injury, such as esophageal perforation or intramural dissection, may occur, albeit rare, especially in patients complaining of severe pharyngeal and chest pain [21,22]. Since the condition may be exacerbated by performing urgent endoscopy in such patients, computed tomography (CT) and esophagogram should be considered before endoscopy.

## 3. Disease Concept and Endoscopic Abnormalities

The disease entity of EoE has been distinctly established by the report of Attwood et al. in 1993, in which they described a case series of 12 patients characterized by dysphagia as a chief complaint, young male predominance, no pathological gastro-esophageal reflux, intense, eosinophilic inflammation localized in the esophagus, and normal endoscopy [23]. However, a subsequent large number of endoscopic studies have revealed that EoE displays several non-specific, but abnormal findings [24,25,26]. In many studies around the 1990s to early 2000s, those endoscopic findings have been described by terms as varied as the following: ringed/corrugated esophagus, concentric/fixed rings, esophageal trachealization, feline esophagus, linear/longitudinal furrows, linear fissures, white plaques/exudates, white stipple-like exudates, cobblestone-like pattern, decreased/reduced vascularity, loss of vascularity, edema, stricture/stenosis, narrow/small caliber esophagus, long segment strictures, mucosal fragility, and crepe-paper esophagus [24,25,26,27,28,29,30,31,32,33].

In a previous meta-analysis, EoE was reported to be endoscopically “normal” in 20% of retrospective studies, but the ratio decreased to only 7% in prospective studies [7]. In other words, more than 90% of EoE have no endoscopic abnormalities. However, the diagnostic operating characteristics of each finding have been shown to be unsatisfactory, with a low sensitivity of 15–67% (edema 54%, rings 67%, exudates 47%, furrows 55%, stricture 15%) and a low positive predictive value of 27–60% (edema 49%, rings 60%, exudates 60%, furrows 60%, stricture 27%) against a high specificity of 82–93% (edema 82%, rings 91%, exudates 93%, furrows 93%, stricture 93%) and high negative predictive value of 85–93% (edema 85%, rings 93%, exudates 89%, furrows 92%, stricture 86%) when analyzed in the prospective studies. In contrast, when analyzing the presence of at least one of those five major findings, sensitivity (91%) and negative predictive value (97%) increased, but specificity (65%) and positive predictive value (39%) decreased. Up to 100 reports analyzed in this meta-analysis have been reported from 1985 to 2011 prior to the publication of the EREFS scoring system; therefore, the assessment of endoscopic abnormalities varies considerably among the studies. The lack of objective and unified terms representing endoscopic abnormalities is deemed to have impacted the diagnostic operating characteristics, in addition to the low awareness of disease, low resolution of endoscopy, disease prevalence, and different settings in which endoscopy was performed.

## 4. EoE Endoscopic Reference Score (EREFS)

The EoE endoscopic reference score (EREFS) was developed by Hirano et al. in 2013 for objective assessment of endoscopic findings: Edema, Rings, Exudates, Furrows, and Stricture are comprehensively assessed as major features [8]. Soon after, its adjusted version by van Rhijn et al. was reported in 2014 to be useful not only for expert but also for non-expert endoscopists [34] (Figure 2).

The EREFS acronym consists of the initials of the aforementioned five endoscopic findings. Since the first E (Edema) and second E (Exudates) may be confused, the term ERExFS may be preferred [35]. Ma et al. compared the original, simplified, expanded furrows and fully expanded EREFS classification, using paired endoscopic videos from patients with EoE before and after treatment [36]. The original EREFS, the simplified EREFS, and the fully expanded EREFS are totally scored with ranges of 0 to 8, 0 to 7, and 0 to 11, respectively (Table 1). When the distribution and the extent of endoscopic abnormalities are heterogeneous within the esophagus, it is common for EREFS scores to be assessed in the most affected area [19].

### 4.1. Major Features of EREFS

Edema, used synonymously with loss of or decreased vascularity, is quite an unspecific finding seen in many other esophageal conditions, including GERD, and has endoscopically lower diagnostic agreement compared with other findings [8,34].

Rings occur as a result of esophageal remodeling with subepithelial fibrosis as well stricture, and correlate with a risk of food impaction attributed to impaired esophageal distensibility [37]. The severity of rings does not correlate with the number of infiltrating eosinophils, and tends to remain after histologic improvement [38,39]. It is known that rings are also observed in up to 10% of non-EoE esophagitis, including GERD [30,40,41] (Figure 3).

Exudates appear as white plaques or white spots, corresponding to histological eosinophilic microabscesses [42]. Attention should be paid, in the assessment of exudates, to the coexistence of other EoE characteristic findings for the differential diagnosis with candidiasis, as both conditions are typical of dysphagia [43].

Furrows are crack-like fissures longitudinally running in the esophagus, different from the epithelial erosions seen in GERD (Figure 3). Furrows are more easily recognized by decreasing the tension of the esophageal lumen with air deflation, or by employing indigo carmine spray or by blood falling into the fissure after biopsy [44] (Figure 3). EoE and erosive esophagitis may coexist due to their interactional pathophysiology, and EoE is diagnosed if esophageal eosinophilia with a peak of ≥15 eosinophils/hpf is present on biopsies [45].

Stricture with EoE, unlike that in GERD, can occur at a more proximal site of the esophagus [30,32,46]. In clinical practice, the appearance of a stricture may be almost synonymous with the narrow/small caliber esophagus described below.

Considering their relevant pathophysiology, rings and stricture are separately assessed as fibrostenotic changes, apart from the remaining three inflammatory changes edema, exudates, and furrows. [46,47].

The optimal cutoff value for the ERERS score to predict EoE has not been determined. Some reports showed that an EREFS ≥ 2 can predict EoE with relatively high sensitivity and specificity [38,48], while others suggested that it was not sufficient to discriminate between EoE and non-EoE, due to low specificity against high sensitivity [49]. This discrepancy may be due to differences in the endoscopic severity of the EoE group enrolled in each study, and the clinical profiles of the non-EoE comparative groups. Endoscopists should also be aware that endoscopic disease activity by EREFS score does not correlate well with clinical symptoms or histological activity [50,51].

### 4.2. Minor Features of EREFS

Feline esophagus, narrow caliber esophagus, and crepe paper esophagus were considered in the EREFS as minor features [8].

Feline esophagus is a term derived from the morphological similarity to a cat’s esophagus, in which multiple rings are physiologically present. However, this term now refers to fine and subtle contractile rings observed transiently or intermittently during endoscopy in non-EoE conditions, including normal esophagus [8,52,53] (Figure 3).

Narrow/small caliber esophagus indicates the diffuse narrowing of the tubular esophagus with a certain longitudinal length, which is different from the strictures seen in GERD or in postoperative anastomotic stricture, presenting as shorter, localized narrowing along with proximal esophageal dilatation. These two conditions are not clearly defined or differentiated from each other [54,55]. Carlson et al. proposed an arbitrary definition: narrow caliber esophagus is defined as an esophageal diameter < 18 mm for > 50% of the length of the organ [55]. Endoscopy had a low sensitivity of 25% in detecting esophageal narrowing, with a radiographic maximal diameter of ≤18 mm [54]. Endoscopists need to know that narrow caliber esophagus is less successfully assessed by endoscopy, compared with a barium esophagogram [56,57] (Figure 2 and Figure 4).

Crepe-paper esophagus is an uncommon but pathognomonic finding with EoE, presenting with friable, delicate esophageal mucosa which ulcerates after minor trauma, such as by contact with an endoscope (Figure 4). Of those three minor features, crepe-paper esophagus had the highest diagnostic agreement among endoscopists, and was eventually adopted as a minor feature [8].

## 5. Diagnostic Agreement of Endoscopic Findings and Diagnosis

In EoE, diagnostic delay without effective therapeutic intervention not only exacerbates the degree of esophageal stricture [58], but also leads to impaired quality of life [59], decreased treatment response [60], repeated esophageal dilatation [61], and an increased risk of critical mechanical esophageal injuries, such as perforation [21] and intramural esophageal dissection [22]. Hence, it is critical to diagnose and treat EoE in the earlier phase [62]. To this end, it is crucial to identify endoscopic abnormalities with a high degree of diagnostic agreement, as a trigger for the endoscopist to perform a biopsy.

Previously reported endoscopic diagnostic agreement of individual components of EREFS, and the overall diagnosis of EoE, are summarized in Table 2. Most studies have focused on EoE images using white light, and a few studies have included non-EoE images in the diagnosis of EoE. Although the results cannot be directly compared because of the differences in observers or image evaluation methods among studies, many reports showed relatively good interobserver agreement regarding furrows and rings with white light imaging (WLI): furrows, moderate to excellent (0.48–0.68); rings, fair to excellent (0.34–0.70). In contrast, interobserver agreement was fair to moderate (0.24–0.5) for edema, fair to substantial (0.21–0.65) for exudates, and fair to moderate (0.31–0.5) for stricture; thus, it was lower than that for furrows and rings, with a variation among the studies. The intraobserver agreement in WLI was generally good, ranging from moderate to substantial (0.43–0.78), except for exudates; they were fair (0.28) in a report by Izumi et al. [63]. These authors also showed unsatisfactory agreement for the overall diagnosis of EoE with an interobserver agreement of 0.34, and an intraobserver agreement of 0.52 in WLI. In a more recent Japanese study, interobserver and intraobserver agreement have been reported to increase to 0.60 and 0.74 for WLI, respectively. This might reflect an increasing awareness of EoE [48]. The comparison between original EREFS and expanded EREFS by Ma et al. does not show a significant difference in the diagnostic agreement regarding each endoscopic finding [36].

Over the past decade, image enhanced endoscopy (IEE) such as narrow band imaging (NBI) has been widely used in endoscopic diagnosis, especially for gastrointestinal cancerous lesions [64,65,66]. There have been a few reports investigating the usefulness of IEE for the diagnosis of EoE. Prior to the publication of the EREFS scoring system, Peery et al. reported that NBI had no additional effect on WLI in the diagnosis of EoE, when only endoscopists assessments of WLI images were compared with WLI, plus its counterpart NBI images [67]. Gastroenterologists who participated in that study reported that 50% of the images were better viewed with NBI, 7% of the images were better viewed with WLI, and 43% of the images were undecided on. In other reports using NBI combined with magnifying endoscopy, the esophageal mucosa in patients with EoE had a beige color, increases in dot-like intrapapillary capillary loops, and invisibility of cyan submucosal vessels [68,69] (Figure 5). The beige-colored mucosa under NBI has also been found to correspond to an area with histologically active eosinophilic inflammation [70].

LCI is a newly-developed IEE created by short-wavelength narrow-band laser light combined with white laser light, enabling brighter light in a distant area, and enhancing color differences between red and white [71]. It has been increasingly reported that LCI can improve various esophageal diseases, such as minimal change esophagitis [72,73], erosive esophagitis [74], Barrett’s esophagus, and esophageal adenocarcinoma [75,76,77]. There is a brief report showing that LCI could effectively detect an area of active inflammation, which was observed as yellowish mucosa, compared with cyan or light purple mucosa without active inflammation [78] (Figure 5). We also recently showed that LCI in addition to WLI improved the diagnostic accuracy of the individual endoscopic findings, especially furrows, rings, and stricture, and the overall diagnosis of EoE with “moderate” to “substantial” consistency compared with WLI alone. The improvement in diagnostic accuracy by LCI was more remarkable in cases with milder endoscopic findings than in those with more prominent findings, and in endoscopists with less EoE experience than in those with more EoE experience [48]. It is speculated that LCI potentially improves the visibility of the EoE by increasing the contrast between red and white areas, and illuminating more brightly up to the more distal esophagus (Figure 6). Further investigations are needed to determine whether IEE can improve endoscopic diagnostic agreement for EoE.

## 6. Other Endoscopic Findings Suggestive of EoE

With an increased awareness of EoE in clinical practice, endoscopic findings suggestive of EoE, other than the characteristic features such as EREFS, have been known to endoscopists. These findings are summarized in Table 3, and their representative images are shown in Figure 7. Several findings, such as multiple polypoid lesions [79,80,81], tug sign/pull sign [82,83], ankylosaurus back sign [84], and caterpillar sign [49] have been reported. In addition, endoscopic abnormalities have been localized in small areas of the lower end of the esophagus, or with patchy distribution in 30–40% of patients [85,86] (Figure 8). Such localized types may be missed, unless the whole esophagus is carefully observed with proper expansion by both air inflation and deep inspiration during endoscopy. Dynamic observation with control of the extensibility of the esophageal wall may help an endoscopist detect milder and localized endoscopic abnormalities (Figure 8). Localized EoE appears to have less symptoms and higher responsiveness to PPI therapy compared with diffuse EoE [87,88], but its etiology and potential progression to diffuse EoE remain to be elucidated. The American Gastroenterological Association has proposed to incorporate into the severity scoring of EoE (in addition to symptoms, complications, and histological features) whether inflammatory features are localized or diffuse, as well as the grade of fibrostenotic features measured by the ability to pass a normal diameter upper endoscope [89].

## 7. Other New Modalities for Endoscopic Diagnosis

It has been recently reported that endocytoscopy, using super-magnifying technology, can microscopically detect infiltrated intraepithelial eosinophils, characterized by bilobed nuclei [90,91]. Endocytoscopy has not yet been commonly used in the clinical setting owing to the complicated procedures including cell staining and the related cost. Moreover, it can only provide information on cellular and mucosal structures in a very superficial area of the esophagus. However, it may serve as a promising tool for more effectively detecting the esophageal site with abundant eosinophils, and thus decreasing the invasiveness of multiple biopsies.

Artificial intelligence (AI) can endoscopically discriminate EoE from normal esophagus with > 95% accuracy, higher than that of the endoscopist, leading to decreasing diagnostic delay by recommending re-endoscopy with esophageal biopsies, or consultation with a specialist for EoE—especially in cases with non-expert endoscopists [92,93]. Although still images have been mainly used in previous studies, real-time diagnostic assistance during endoscopy, for encouraging biopsies, may become possible if the diagnostic algorithm using video learning is established in the future. AI may also predict non-invasive histological remission with high accuracy, as reported in ulcerative colitis [94,95]. However, some issues remain to be improved in AI diagnosis. In a Japanese report, AI-based diagnosis tended to misdiagnose milder endoscopic abnormalities as normal esophagus (a factor of low sensitivity), and to misdiagnose normal structures such as vertical lines, transient ring or glycogenesis acanthosis as EoE (a factor of low specificity), similar to what was also conducted by endoscopists [93]. Especially in Japan, milder patients with both symptoms and endoscopic findings have been commonly diagnosed in the GI screening for health checkups [96,97]. Despite the recent increasing recognition of endoscopic abnormalities with EoE, it has been shown that up to 10% of EoE patients have a normal-appearing esophagus [98]. In real practice, the endoscopist faces various esophageal diseases and conditions such as erosive esophagitis, esophageal Candida, and neoplastic lesions. Whether high diagnostic accuracy can be maintained even after learning numerous images—that include typical EoE, mild EoE, normal-appearing EoE and other diseases, including those endoscopically similar to EoE—is a future challenge [99]. Eventually, the endoscopist needs to record clear and detailed images for AI to operate on with greater precision.

## 8. Esophageal Biopsy and Histological Assessment

Multiple biopsies are favorable to diagnose EoE more effectively, due to a heterogeneous distribution of intraepithelial eosinophil infiltration [100,101,102]. It is recommended that ≥2–4 biopsies by ACG (American College of Gastroenterology) guidelines [11] and ≥6 biopsies by UEG (United European Gastroenterology) guidelines [10] should be obtained from more than two different locations. Distribution of eosinophils varies among different endoscopic findings and locations. Furrows and exudates containing more intense eosinophil infiltrations should be preferentially targeted in biopsies [31,96,101]. The biopsy should be selectively obtained from just above the furrows located in valleys, rather than from the mucosa between the valleys, as significantly higher eosinophils are detected in the former [103]. Lines with subtle linear appearances without cracks are not an appropriate target for biopsy due to less eosinophilic infiltration [101]. Rings or stricture, which reflect subepithelial fibrosis, are less favorable targets for biopsy due to a poor correlation with active eosinophilic inflammation [39,51,101]. Histological assessment should include biopsy samples from the distal esophagus, which is more heavily infiltrated with eosinophils than the proximal esophagus [101,104]. Once again, up to 10% of patients exist with an endoscopically normal esophagus [98]. Thus, when EoE is clinically suspected, the endoscopist should obtain esophageal biopsies even if no apparent endoscopic abnormalities are detected. As noted above, endoscopy should be conducted without the use of PPIs whenever possible to avoid underdiagnosis with endoscopic and histologic remission by PPI [16,20].

A peak eosinophil count with ≥15 eosinophils/hpf has been conventionally used as a reliable histological criterion for the definitive diagnosis in EoE [105], which generally indicates an increase in intraepithelial eosinophils. Schoepfer et al. reported that peak intraepithelial eosinophil counts were higher in 63%, identical in 7%, and lower in 30% of patients, compared with the subepithelial layer [106]. Forty percent of EoE patients with <15 intraepithelial eosinophils/hpf had subepithelial peak counts of ≥15, which were thus not diagnosed as EoE, unless biopsy samples included adequate subepithelial tissue. In addition, eosinophilic esophageal myositis (EoEM) with predominant eosinophil infiltration in the muscle layer has been proposed as a subtype of EGIDs, suggesting an association with esophageal motor disorders such as Jackhammer esophagus [107]. Neither endoscopic abnormalities, such as typical EoE, nor increased intraepithelial eosinophils have been found in this condition. For definitive diagnosis, endoscopic ultrasonography-guided fine needle aspiration biopsies and per oral esophageal muscle biopsies need to be performed [94,95]. Thus, when histologically evaluating EoE based on eosinophil count in mucosal biopsies, it is important to note the heterogeneity of eosinophil distribution by biopsy specimen or by horizontal and vertical segments within the esophagus, the diversity of eosinophil counting methods (peak/average number of eosinophils and number of fields of view measured), and the differences in the microscopes used at each institution [108].

Besides absolute counts of eosinophils, EoE has various histological features such as eosinophilic microabscesses, surface laying of eosinophils, eosinophil degranulation, basal cell hyperplasia, lengthening of lamina propria papillae, increased intraepithelial lymphocytes and mast cells, increased intercellular edema, and increased lamina propria fibrosis [42]. Recently, the EoE histologic scoring system (EoE-HSS), which comprehensively evaluates those histological findings including their severity and extent, has been proposed as a more accurate assessment for disease activity or differentiation from GERD [109,110]. Although it has been expected to be useful especially in clinical research or pharmacological trials, it may be too complicated to be used in the real clinical setting, at least in its current version. The more simplified EoE-HSS is desirable. Eosinophil quantification using AI has been recently shown to be helpful in overcoming several challenges including heterogeneous distribution of eosinophils, uncertainty of tissue sampling, or problems with biopsy specimen processing, such as cases where specimens are not cut out vertically [111].

In the absence of gastroenteritis-like symptoms or apparent endoscopic abnormalities in the stomach or the duodenum, gastroduodenal biopsies seem not to be necessary, at least in adult patients because esophageal involvement of eosinophilic gastroenteritis is unlikely [112,113].

## 9. Conclusions

A variety of factors—differences in endoscopists’ interests and experience, differences in endoscopic equipment performance, the presence of cases with weak or normal appearing esophagus, and the variability in eosinophil infiltration—have influenced diagnostic accuracy in EoE. However, increasing awareness of EoE and its clinical guidelines have led to a decrease in diagnostic delay, endoscopic severity at first diagnosis, and number of endoscopic examinations performed before diagnosis, and consequently more cases are being diagnosed at an earlier stage [114]. In addition to the recent spread of high-resolution endoscopy with IEE, establishing the EREFS scoring system has contributed to the improvement in the diagnostic accuracy of EoE. Newly developed technology, such as endocytoscopy or AI, may support more robust and less invasive definitive diagnosis and management of EoE in real clinical practice.

## Figures and Tables

**Figure 1 diagnostics-12-03202-f001:**
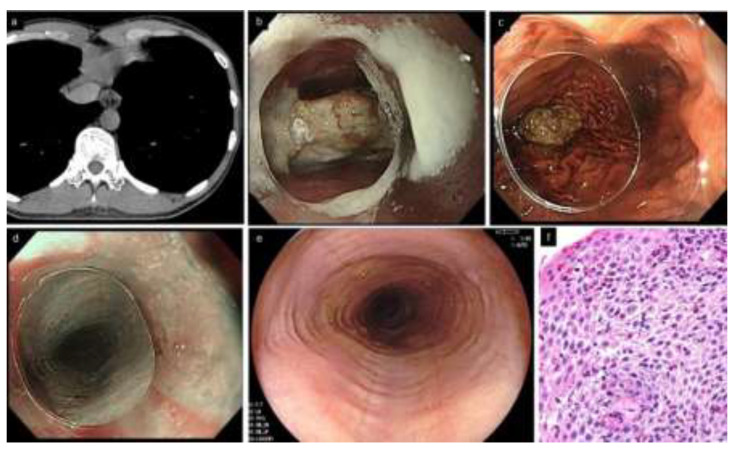
A typical patient with EoE complicated by food impaction (33 years old, male). (**a**). Chest CT. Esophageal lumen is dilated and filled with structures that appear to be impacted food. (**b**,**c**). Urgent endoscopy. Food bolus (piece of meat) is present in the middle esophagus. Impacted food is removed by letting it drop into the stomach, pushing with an endoscope equipped with a clear distal attachment hood. (**d**). Subtle rings and furrows are visible by narrow band imaging, suspicious of EoE. (**e**). Rings and furrows are more distinctly observed in the re-endoscopy after one week without PPI. (**f**). Esophageal biopsies show a peak of 52 eosinophils/hpf. EoE, eosinophilic esophagitis; PPI, proton pump inhibitor.

**Figure 2 diagnostics-12-03202-f002:**
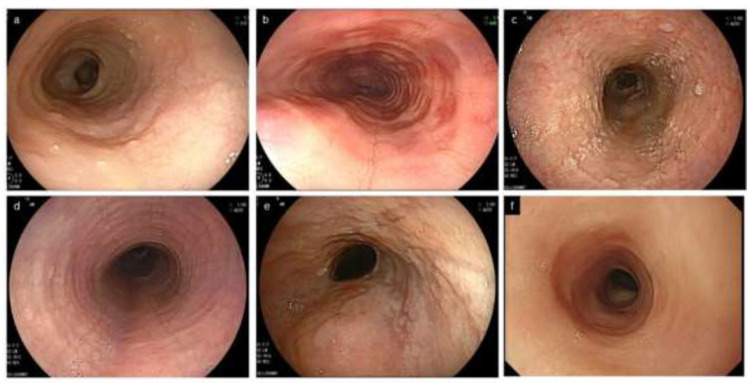
Characteristic endoscopic findings of EoE. (**a**). Edema. Decreased vascularity or loss of it is evident. (**b**). Rings. Multiple concentric rings are almost constitutively present (esophageal trachealization). (**c**). White exudates. White plaques, difficult to distinguish from Candida, are present. (**d**). Furrows. Multiple crack-like lines running longitudinally are presented. (**e**). Stricture. Web-like appearance with proximal esophageal dilatation is displayed. (**f**). Narrow caliber esophagus. Esophageal narrowing with tubular-appearing lumen is demonstrated. EoE, eosinophilic esophagitis.

**Figure 3 diagnostics-12-03202-f003:**
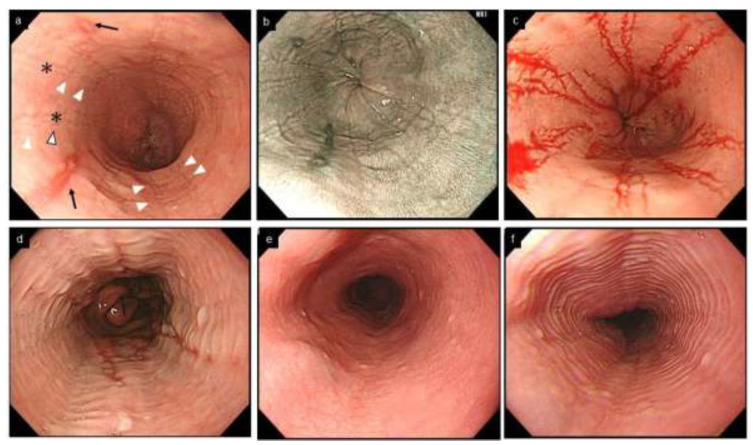
Esophageal erosion, furrows and rings in GERD and EoE, and feline esophagus in normal esophagus. (**a**,**b**). Coexistence of furrows in EoE (arrowhead) and longitudinal erosion in GERD (black arrow). Biopsy should be obtained just above on furrows (asterisk). (**c**). The visibility of furrows is enhanced after esophageal biopsies with blood pouring on the furrows. (**a**–**c**, same patient; **a**, WLI; **b**, NBI). (**d**). Mild rings observed in erosive esophagitis. (**e**,**f**). Feline esophagus. Transient and subtle concentric rings are observed in the normal esophagus (**e**,**f**, same patient). EoE, eosinophilic esophagitis; GERD, gastroesophageal reflux disease; WLI, white light imaging; NBI, narrow band imaging.

**Figure 4 diagnostics-12-03202-f004:**
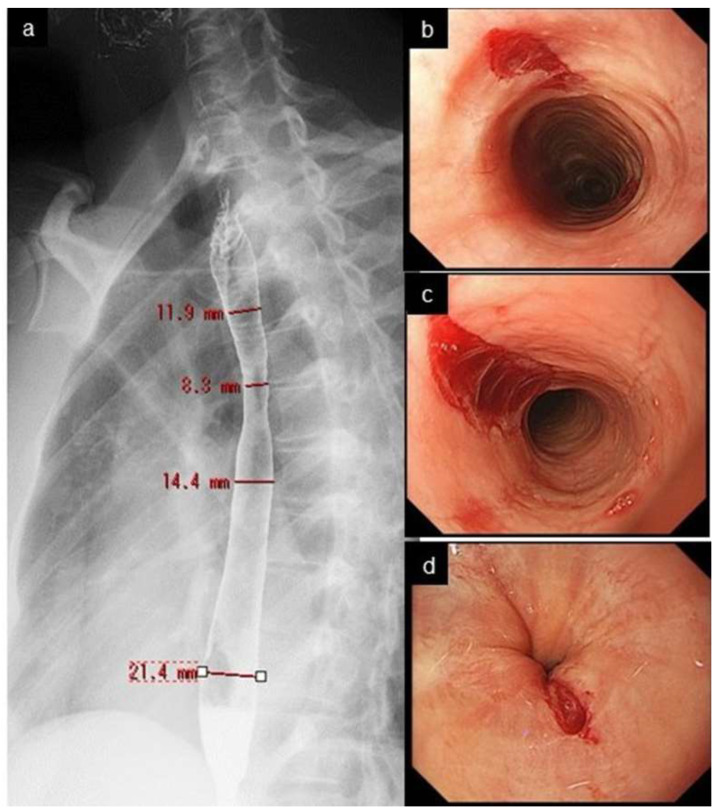
Narrow caliber esophagus and crepe-paper esophagus. Esophagogram shows narrow caliber esophagus from the upper to mid esophagus, with a diameter of 8.3–14.4 mm. (**a**). Mucosal tear developed at upper (**b**), mid (**c**), and lower end (**d**) of the esophagus after regular passage of peroral endoscope.

**Figure 5 diagnostics-12-03202-f005:**
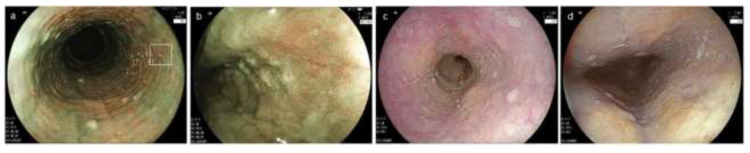
Endoscopic findings by IEE in EoE. (**a**). Beige color mucosa (BLI) (**b**). Magnified image of the area indicated by the white frame in Figure (**a**). Dot-like intrapapillary capillary loops are seen. Cyan submucosal vessels are invisible (BLI). (**c**). Yellowish mucosa (LCI, proximal esophagus). (**d**). Yellowish mucosa (LCI, middle esophagus). IEE, image enhanced endoscopy; EoE, eosinophilic esophagitis; BLI, blue laser imaging; LCI, linked color imaging.

**Figure 6 diagnostics-12-03202-f006:**
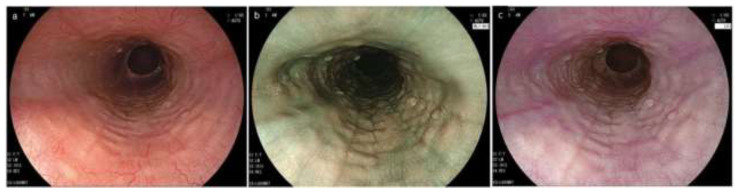
Endoscopic image sets consisting of WLI, BLI and LCI in EoE. (**a**) WLI; (**b**) BLI; (**c**) LCI (**a**–**c**, same patient)**.** Rings and edema in the lower esophagus are more clearly visible in LCI compared to WLI and BLI. EoE, eosinophilic esophagitis; WLI, white light imaging; NBI, narrow band imaging; BLI, blue laser image; LCI, linked color imaging.

**Figure 7 diagnostics-12-03202-f007:**
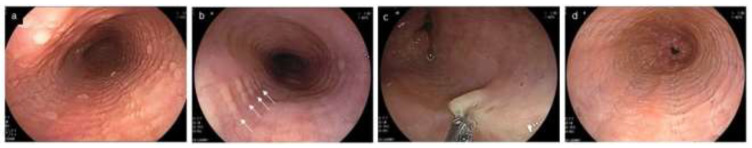
Other endoscopic findings related to EoE. (**a**). Multiple polypoid lesions (**b**). Ankylosaurus back sign (**c**). Tug sign/pull sign: with pulling by the biopsy forceps, a large amount of esophageal mucosa is lifted as a tent and caught in the forceps opening. (**d**). Caterpillar sign. EoE, eosinophilic esophagitis.

**Figure 8 diagnostics-12-03202-f008:**
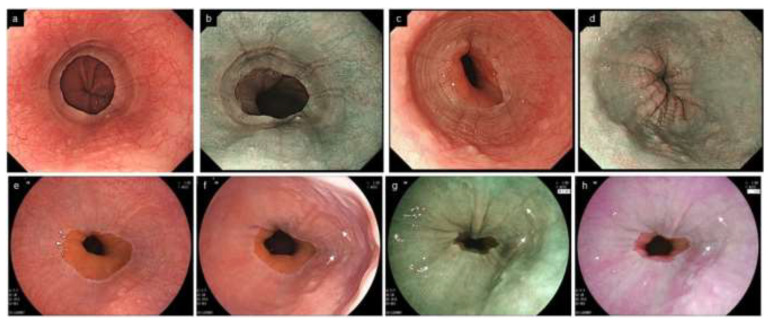
Localized EoE in the lower end of the esophagus. Endoscopic abnormalities are localized at a small area of 1–2 cm in the lower end of the esophagus. (**a**,**b**). Furrows and edema are present in the lower end of the esophagus (same patient, **a**, WLI; **b**, NBI). (**c**,**d**). Furrows, rings and edema in the lower end of the esophagus (same patient, **c**, WLI; **d**, NBI). (**e**). No abnormal findings appear to be found under full expansion of the esophageal lumen (WLI). (**f**–**h**). Reducing the esophageal wall tension increases the visibility of furrows (arrow) and edema. Dynamic observation controlling the extensibility of the esophageal wall increases the visibility of furrows and edema (same patient, **f**, WLI; **g**, BLI; **h**, LCI). EoE, eosinophilic esophagitis; WLI, white light imaging; NBI, narrow band imaging; BLI, blue laser imaging; LCI, linked color imaging.

**Table 1 diagnostics-12-03202-t001:** The EoE endoscopic reference score (EREFS).

Endoscopic Findings	Grade	Hirano et al. [8]	van Rhijn et al. [34]	Ma et al. [36] *
Edema	absent (distinct vascularity is present)	Grade 0	Grade 0	Grade 0
	mild (reduced vascularity or loss of clarity of vascular markings)	Grade 1		Grade 1
	severe (absence of vascular markings)		Grade 1	Grade 2
Rings	none	Grade 0	Grade 0	Grade 0
	mild (subtle circumferential ridges)	Grade 1	Grade 1	Grade 1
	moderate (distinct rings that do not impair the passage of a standard diagnostic adult endoscope [outer diameter 8–9.5 mm])	Grade 2	Grade 2	Grade 2
	severe (distinct rings that do not permit the passage of a diagnostic endoscope)	Grade 3	Grade 3	Grade 3
Exudates	none	Grade 0	Grade 0	Grade 0
	mild (lesions involving < 10% of the esophageal surface area)	Grade 1		Grade 1
	moderate (lesions involving > 10% and <25% of the esophageal surface area)	Grade 2	Grade 1	Grade 2
	severe (lesions involving > 25% of the esophageal surface area)			Grade 3
Furrows	absent	Grade 0	Grade 0	Grade 0
	mild (vertical lines present without visible depth)	Grade 1	Grade 1	Grade 1
	severe (vertical lines present with mucosal depth [indentation])			Grade 2
Stricture	absent	Grade 0	Grade 0	Grade 0
	present	Grade 1	Grade 1	Grade 1

* Fully expanded EREFS.

**Table 2 diagnostics-12-03202-t002:** Inter- and intraobserver agreement in the endoscopic diagnosis for EoE.

Authors	Image Evaluation	Rater	Mode	Inter/intra Observer Agreement	Kappa Value					
					Edema	Rings	Exudates	Furrows	Stricture	Dx of EoE
**Peery et al. [67]**	dichotomous definition	Ex 35, non-Ex 42	WLI	inter	-	0.56	0.29	0.48	-	-
			WLI+NBI	inter	-	0.5	0.24	0.49	-	-
			WLI	intra *	-	53%	50%	69%		-
**Hirano et al. [8]**	EREFS	Ex 7, non-Ex 14	WLI	inter	0.43	0.4	0.46	0.54	0.52	-
				intra	-	-	-	-	-	-
**van Rhijn et al. [34]**	adjusted EREFS ^#^	Ex 4, non-Ex 4	WLI	inter	0.24	0.7	0.65	0.49	0.54	-
				intra	0.49	0.64	0.69	0.69	0.54	-
**Izumi et al. [63]**	dichotomous definition	Ex 20, non-Ex 20	WLI	inter	0.26	0.34	0.21	0.48	-	0.34
				intra	0.43	0.51	0.28	0.55	-	0.52
**Ma et al. [36]**	original, simplified, expanded EREFS ^†^	Ex 15	WLI	inter (original)	0.5	0.69	0.51	0.54	0.39	-
				inter (fully expanded)	0.42	0.69	0.41	0.54	0.39	-
				intra (original)	0.64	0.74	0.78	0.58	0.67	-
				intra (fully expanded)	0.67	0.74	0.6	0.69	0.67	-
**Abe et al. [48]**	adjusted EREFS	Ex 4, non-Ex 6	WLI	inter ^‡^	0.36	0.53	0.34	0.68	0.31	0.6
			WLI+LCI	inter ^‡^	0.35	0.5	0.32	0.74	0.5	0.7
			WLI	intra ^‡^	0.51	0.45	0.47	0.78	0.55	0.74
			WLI+LCI	intra ^‡^	0.49	0.64	0.57	0.85	0.59	0.83

* The proportion of participants having kappa value of fair to good (0.40–0.75). # In the adjusted EREFS, the grading of edema and exudates was modified and simplified, respectively (see Table 1). ^†^ In the fully expanded version, the grading of edema (0–2), exudates (0–3), and furrows (0–2) was expanded (see Table 1). The grading of rings and stricture in the fully expanded version is the same as in the original version. ^‡^ The results are shown for diffuse type EoE. EREFS, EoE endoscopic reference score; Ex, expert; non-Ex, non-expert; WLI, white light image; LCI, linked color imaging; Dx, diagnosis.

**Table 3 diagnostics-12-03202-t003:** Characteristics, potential pathology and clinical significance of other endoscopic findings related to EoE.

Endoscopic Findings	Characteristics	Potential Pathology/Clinical Significance	References
Multiple polypoid lesions	Small and multiple lesions resembling esophageal papilloma or glycogenic acanthosis	Unknown	[79,80,81]
Tug sign/pull sign	Stronger resistance than expected while obtaining biopsy sample	More common in PPI responders than non-responders Esophageal remodeling with subepithelial fibrosis	[82,83]
Ankylosaurus back sign	Multiple small nodules running longitudinally on esophageal mucosa ridges	More common in PPI responders (especially accompanied by erosive esophagitis) than PPI non-responders	[84]
Caterpillar sign	Caterpillar tracks or runway-like marks formed by longitudinal furrows and concentric rings	Higher diagnostic accuracy than EREFS score ≥ 2	[49]

PPI, proton pump inhibitor; EREFS, EoE endoscopic reference score.

## Data Availability

Not applicable.

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
