# Peer review of "Endoscopic Diagnosis of Eosinophilic Esophagitis: Basics and Recent Advances"

_diagnostics, 2022, doi:10.3390/diagnostics12123202_

Round 1

Reviewer 1 Report (Previous Reviewer 2)

I have checked the author's reply and have found that my comments have been addressed properly. I have no further comments or suggestions. 

Reviewer 2 Report (Previous Reviewer 3)

The revised manuscript has a lot of corrections in the English language and clarity of contents. I would like to suggest the publication of this impressive article which is of great help. I would also like to congratulate the authors to put in an effort to bring this work together.

This manuscript is a resubmission of an earlier submission. The following is a list of the peer review reports and author responses from that submission.

Round 1

Reviewer 1 Report

Eosinophilic esophagitis (EoE) is a chronic immune-mediated inflammatory disease characterized by esophageal dysfunction and intense localized eosinophil infiltration in the esophagus, which has raised widespread concern as its incidence continues to rise. Based on literature search and analysis, the authors review the research progress of endoscopy and biopsy techniques in EoE, as well as discuss the frontier development of optical techniques. This is an intriguing topic for the researchers involved. However, the paper needs very significant improvement before acceptance for publication.

Major comments:

1. The current manuscript’s logical relationships are confusing, making it difficult for readers to understand. The authors need to reorganize the manuscript, think about the logical relationships within and between paragraphs, delete the descriptions that are not strongly related or even unnecessary, and revise the use of related words to make the manuscript more organized and clearer. Take “1. Introduction” as an example, the authors should briefly describe the current status, incidence, problems and difficulties of EoE in this part, and then introduce the authors' viewpoint to pave the way for further elaboration below.

2. Literature evidence should be reasonably summarized and analyzed when cited, and further elaborated or extended appropriately combined with the topic to expand the depth of the manuscript, rather than simply describing the relevant results and data, lacking reasonable language organization and inconveniencing the reader.

3. The authors should highlight in the manuscript the innovation and importance of this research for disciplinary advances and clinical applications in related fields.

4. The language of the full text should be critically revised, including grammar and headings, to make the manuscript more concise and fluent.

Minor comments:

1. Page 1, Line 29, Please write the full name in the body of the manuscript at the first time using the abbreviation.

2. Page 1, Line 42, Please revise the reference format according to the journal's requirements.

3. Page 2, Line 58, What kind of patients does the term "such patients" refer to?

4. Please make the markings in Figure 3 more visible to the readers.

5. The authors should make the tables more attractive and concise by adjusting the font size and typography. In addition, please revise the lines in Table 1; ensure that the symbols in legend of Table 2 match the table; put legend of Table 3 off the table; please list the full names of all abbreviations in the legend.

Reviewer 2 Report

Thanks for inviting me to review this paper. Overall this is a well organized and beautifully presented work and I read it with great interest. The authors should be congratulated for their achievement. I have some comments. First, I suggest authors to supplement data about sensitivity and specificity of endoscopic features in the diagnosis of EoE, at least including major characters such as edema, rings, exudates, furrows, and stricture. Secondly, it is a surprise to note that the inter- and intra-observer agreement in the endoscopic diagnosis of EoE is quite low. The authors should comment on this and provide suggestions to improve the inter- and intra-observer agreement. Thirdly,  the limitation of current endoscopic technique to diagnose EoE needs be addressed. BTW, Line 294 should be blue laser, not blue lase.

Reviewer 3 Report

I would suggest a minor briefing due to the length of the article where the number of words can be reduced but overall I liked the work. It was a helpful article for me as a GI provider. I suggest publication.